# A Novel Fault Detection Method Based on One-Dimension Convolutional Adversarial Autoencoder (1DAAE)

**Jian Wang \*, Yakun Li and Zhiyan Han**

School of Control Science and Engineering, Bohai University, Jinzhou 121013, China
\*   Correspondence: ganard@163.com

**Abstract:** Fault detection is an important and demanding problem in industry. Recently, many researchers have addressed the use of deep learning architectures for fault detection applications such as an autoencoder. Traditional methods based on an autoencoder usually complete fault detection by comparing reconstruction errors, and ignore a lot of useful information about the distribution of latent variables. To deal with this problem, this paper proposes a novel unsupervised fault detection method named one-dimension convolutional adversarial autoencoder (1DAAE), which introduces two new ideas: one-dimension convolution layers for the encoder to obtain better features and the adversarial thought to impose the latent variable $z$ to cluster into a prior distribution. The proposed method not only has powerful feature representation ability than the traditional autoencoder, but has also enhanced the discrimination ability by imposing a prior distribution of the latent variables to cluster. Then, two anomaly scores for 1DAAE were proposed to detect fault samples, one based on reconstruction errors, and the other based on latent variable distribution. Finally, it was shown by the experiments that the proposed method outperformed the autoencoder-based, adversarial autoencoder-based, one-dimension convolutional autoencoder-based and generative adversarial network-based algorithms on the Tennessee Eastman process. Through the experiments, we found that the both one-dimension convolution layers and the latent vector distribution are helpful for fault detection.

**Keywords:** fault detection; autoencoder; convolutional layer; Tennessee Eastman process; unsupervised learning

## 1. Introduction

Fault detection is an important and challenging problem in many engineering applications and continues to be an active area of research in the control community such as chemical engineering [1–4], nuclear engineering [5], aerospace engineering [6,7], and automotive systems [8].

Traditional methods for fault detection are mainly based on mechanistic models, which require significant effort and in-depth knowledge to develop. For complex processes, it is not a trivial task to build fundamental models.

For the development of a distributed control system, large volumes of process data that contain valuable information can be well collected and stored effectively. Methods based on data-driven have received more and more attention. In general, the data-driven fault detection methods can be broadly categorized into supervised, semi-supervised, and unsupervised learning methods. The model of supervised and semi-supervised learning needs labeled information to learn a relationship between the input data and the desired output labels. Unsupervised learning does not require fault training data and clusters or classifies data through discovering the powerful features on only the normal data. In practical situations, the fault data cannot easily be collected. Thus, this paper focused on unsupervised learning in fault detection.

The most popular unsupervised learning approaches are principal component analysis (PCA) [9,10] and partial least squares (PLS) [11]. These techniques can project the measurement data from the original high-dimensional space into low-dimensional linear subspace with the covariance or cross-correlation information retained. Then, the fault detection and diagnosis can be performed within the latent variable subspace using Hotelling's T2 (Hotelling's T-squared distribution) and squared prediction error indices. The conventional PCA or PLS monitoring methods are targeted in linear systems, and thus cannot handle nonlinearity in the processes. To deal with nonlinear processes, kernel function-based PCA and PLS approaches have been developed and applied to chemical process monitoring [12]. Basically, kernel PCA or PLS converts the input space into high-dimensional feature space through nonlinear kernel mapping and then the fault detection statistics can be derived from the kernel feature space. In PCA- or PLS-based monitoring methods, the objective is to decorrelate latent variables, therefore, only second-order statistics are taken into account. However, industrial processes are often non-Gaussian, thus, higher-order statistics should not be ignored. More recently, an independent component analysis (ICA)-based monitoring approach has been proposed to tackle non-Gaussian processes [13,14]. The statistically independent latent variables are extracted to track the abnormal operation events in complex processes with significant non-Gaussianity. Though favorable performance may be achieved, the approaches usually require much prior knowledge to hand-design and lack universality.

Recently, deep learning (DL) has been receiving ever-growing attention because of the ability to automatically extract features with multiple levels of abstraction from large amounts of data [15]. In fields such as computer vision, speech recognition, and natural language processing, researchers have been able to build models with much better performances than the traditional models, which require hand-design [16–18]. With the advent of DL, autoencoders (AE) are also used to perform dimension reduction by stacking up layers to form deep autoencoders. By reducing the number of units in the hidden layer, it is expected that the hidden units will extract features that well represent the data. Moreover, by stacking autoencoders, we can apply dimension reduction in a hierarchical manner, obtaining more abstract features in higher hidden layers, leading to a better reconstruction of the data. Many researchers have also addressed the use of autoencoders for faults and anomaly detection applications. In a recent paper, the problem of fault detection has been addressed by building a high level representation of features using a sparse autoencoder (sparse AE) [19]. Additionally, acoustic novelty detection applications have been introduced using AE [20,21]. The AE is trained on the dataset of normal acoustic signals and the novelty is detected if a difference metric between the input and the output of the AE exceeds a threshold. Another interesting work using DL, proposed an adaptive implementation of one-dimension convolutional neural networks (1DCNN) for real-time motor fault detection [22]. Additionally, a DL-based fault diagnosis model has recently been proposed by the extraction of spatial and temporal features using deep belief networks (DBN) [23–26].

Among the above fault detection methods, we found that there was a similar framework. Assuming that we had lower dimensional latent variables $z$, the data $x$ could be represented under the condition of $z$, following the formula $P(x) = P(x|z)P(z)$, where $P(z)$ represents the probability of $z$, $P(x|z)$ represents the model $x$ under parameter $z$ and $P(x)$ represents the probability of $x$. First, these methods try to find the lower dimensional latent variables $z$, which could represent $x$ perfectly. Then, $x$ is brought back to the original data space using $z$, which is called the reconstruction of the original data. By reconstructing the data with low dimension representations, we expect to obtain the true nature of the data, without uninteresting features and noise. The reconstruction error of a data point, which is the error between the original data point and its low dimensional reconstruction, is used to detect faults. The lower dimensional latent variables are the key to reconstructing the original data. The traditional unsupervised algorithms mainly improve the model's fault detection ability by improving the model's feature representation

ability. Though the latent variables have the ability to represent the nature of the original data, there is no guarantee as to where the fault and normal data are separated from each other.

In order to improve this problem, we hope that the latent variable space of AE also satisfies where the fault and normal data could been separated explicitly. We assume that the data $x$ and its label y are independent under the condition of the given lower dimensional latent variable $z$. We present the following generative model:

$$P(x, y|z)P(z) = P(x|z)P(y|z)P(z).$$

where it could be considered as learning a parameterized feature representation model $P(x|z)$ and a discriminant classifier model $P(y|z)$ simultaneously.

Based on the above idea, a novel fault detection method named the one-dimension convolutional adversarial autoencoder (1DAAE) was proposed in this paper. 1DAAE is a multi-layer neural network belonging to a kind of autoencoder, which introduces two new ideas: 1D convolution layers for the encoder to obtain better features and the adversarial thought, which is to impose the latent variable $z$ to cluster into a prior distribution.

The network architectures of 1DAAE include three parts: the encoder, the decoder, and the discriminator for latent variables. As the traditional AE, 1DAAE learns a parameterized feature representation model by the encoder and decoder, which gains richer valuable features by adding one-dimension (1D) convolution layers. Theoretically, any complex latent variable distribution can be mapped to a simple distribution by neural networks, so the discriminator helps the 1DAAE impose the latent variable to follow a prior distribution. Recently, some algorithms with the adversarial thought have been applied in fault detection such as the adversarial autoencoder (AAE) [20] and the generative adversial network (GAN) [27]. The main difference between 1DAAE and the others is that 1DAAE uses 1D convolution layers compared with AAE, and 1DAAE uses an encoder and decoder network structure compared with GAN.

The remaining sections are organized as follows. Section 2 introduces the fault detection based on AE, then describes a novel fault detection algorithm based on 1DAAE. Two abnormal scores are also proposed to enhance the fault detection performance. Section 3 depicts the application of the 1DAAE model on the Tennessee Eastman benchmark and presents the simulation results and their comparison. Finally, we conclude with our conclusions and future work.

## 2. Materials and Methods

### 2.1. Autoencoder

AE is a neural network that is trained by unsupervised learning, which is trained to learn reconstructions that are closed to its original input. AE is composed of two parts, an encoder and a decoder. An encoder and decoder has a single hidden layer, as shown in Equations (1) and (2), respectively. $W$ and $b$ are the weight and bias of the neural network, respectively, and $\sigma$ is the nonlinear transformation function.

$$z = \sigma(W_{xh}x + b_{xh}), \tag{1}$$

$$\tilde{x} = \sigma(W_{hx}z + b_{hx}), \tag{2}$$

$$L_{re} = \|x - \tilde{x}\|, \tag{3}$$

The encoder in Equation (1) maps an input vector $x$ to a hidden representation $z$ by affine mapping following a nonlinearity. The decoder in Equation (2) maps the hidden representation $z$ back to the original input space as a reconstruction. The difference between the original input vector $x$ and the reconstruction $\tilde{x}$ is called the reconstruction error, as shown in Equation (3). AE learns to minimize this reconstruction error. The training algorithm for AE is shown in Algorithm 1.

---

**Algorithm 1:** Autoencoder training algorithm

---

INPUT: Dataset $\quad x_1, \cdots, x_N$

OUTPUT: encoder $f$, decoder $g$

$\quad \omega_f, \omega_g \leftarrow$ Initialize parameters for $f$, $g$

repeat

$L_{re} = \frac{1}{N} \sum_{i=1}^{N} \|x_i - g(f(x_i))\|^2$ Calculate sum of reconstruction error

$\omega_f, \omega_g \leftarrow$ Update parameters using gradients of $L_{re}$

until convergence of parameters $\quad \omega_f, \omega_g$

---

By using the hidden representation of AE as an input to another AE, we can stack the AE to form a deep AE [16]. AEs with various other regularizations have also been developed.

*2.2. Autoencoder Based on Fault Detection Algorithm*

AE-based fault detection is a deviation based fault detection method using unsupervised learning. Usually, it uses the reconstruction error as the anomaly score. Samples with high reconstruction error are considered as fault samples. Only normal samples are used to train the AE. After training, the AE will reconstruct normal data very well, while the fault samples will fail to do so because of the different feather with the normal samples. The fault detection algorithm for AE is shown in Algorithm 2.

---

**Algorithm 2:** AE based on fault detection algorithm

---

INPUT: Normal dataset $X$, Test dataset $x_1, \cdots, x_i, \cdots, x_N$, threshold $\alpha$

OUTPUT: Reconstruction error $\|x_i - \tilde{x}_i\|$

$\omega_f, \omega_g \leftarrow$ Train an AE using the normal dataset $X$

**for** $i = 1$ to $N$

$$L_{re}(i) = \|x_i - g(f(x_i))\|$$

**If** $L_{re}(i) > \alpha$ **then**

$x_i$ is a fault sample

**else**

$x_i$ is a normal sample

**end if**

**end for**

---

*2.3. Proposed 1DAAE Model*

The proposed 1DAAE model consists of one autoencoder (encoder and decoder) and one discriminator. As shown in Figure 1, given an input data $x$, we first encode it as a latent vector $z = f(x)$ using encoder $f$, then the decoder is applied for reconstructing $\tilde{x} = g(z)$. It is important to note that the above steps are by the traditional AE, where we introduce two new ideas: 1D convolution layers, which is shown in Figure 2, for the encoder to obtain better features and the adversarial thought to impose the latent variable $z$ to cluster into a prior distribution.

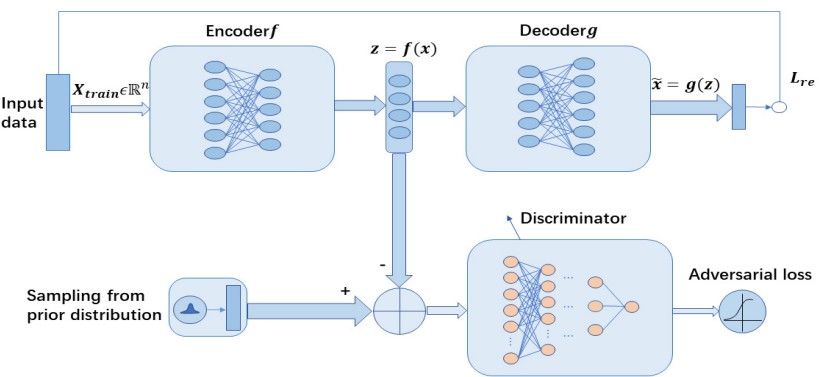

**Figure 1.** Structure of 1DAAE model.

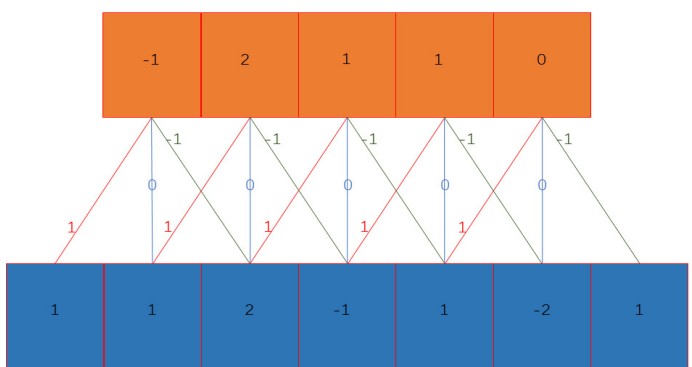

**Figure 2.** The principle of the 1D convolution layer.

### 2.3.1. Network Architecture

As shown in Figure 1, the proposed model is equipped with an encoder–decoder–discriminator structure. The encoder is trained to map the input into an informative latent space, the decoder is used to reconstruct the input data, and the discriminator is to distinguish the data between the prior distribution and latent vector.

- **Encoder Network.** The encoder network comprises a sequence of blocks including different layers: 1D convolution, batch-normalization, and the leaky $ReLU$ activation layer. With the use of the 1D convolutional layers followed by the batch normalization and leaky $ReLU$ activation, respectively, our encoder network is expected to extract useful discriminant features as well as compress the input data to a latent vector $z$. It should be noted that 2D convolution layers have achieved great success in image processing and have extracted a lot of valuable features. Based on the above, the convolutional layers were used in our task to extract more valuable features, but were 1D convolution layers according to our data dimensions. The principle of 1D convolution layer is shown in Figure 2. Suppose the input data are dimension 7 and is convolved by a filter of kernel size 3 (kernel is [1,0,–1]), the corresponding output becomes dimension 5. We can observe the details of the encoder convolution layer in Table 1. The kernel size equipped in the convolutional layers of the encoder is different, $3*3, 5*5, 7*7$, respectively. Given an input sample $x \epsilon X$, the encoder network encodes it as a latent vector $z \epsilon Z$.

**Table 1.** Details of the encoder convolution layer network.

| Name | Type | Filters | Size |
|------|------|---------|------|
| Block1 | Conv | 4 | $3 \times 3$ |
| Block2 | BN | 4 | - |
| Block3 | Conv | 8 | $5 \times 5$ |
| Block4 | BN | 8 | - |
| Block5 | Conv | 16 | $7 \times 7$ |
| Block6 | BN | 16 | - |

- **Decoder Network.** The decoder network usually cooperates with the encoder network to reconstruct the input data from the latent vector $z$. The details of our decoder network are presented in Table 2. First, a linear layer followed by batch normalization was applied to up-scale the latent vector z. Then, a block comprised by the linear layer, batch normalization, and leaky *ReLU* was utilized to up-scale the vector. Eventually, final decoded processing was conducted through a linear layer.

**Table 2.** Details of the decoder network.

| Name | Type | Filters |
|------|------|---------|
| Block1 | Linear + *Leaky ReLU* | 128 |
| Block2 | Linear + BN + *Leaky ReLU* | 256 |
| Block3 | Linear | 52 |

- **Discriminator.** The input of the discriminator is composed of two parts as shown in Table 3: one is the latent vector $z$ generated by the encoder, the other is the data sampled from prior distribution. Two blocks comprising the double linear layer and *Leaky ReLU* activation and one linear following Sigmoid are utilized to discriminate the data.

**Table 3.** Details of the discriminator network.

| Name | Type | Filters |
|------|------|---------|
| Block1 | Linear + *Leaky ReLU* | 128 |
| Block2 | Linear + *Leaky ReLU* | 64 |
| Block3 | Linear | 1 |
| Block4 | Sigmoid | - |

2.3.2. The Training Process

The training process can be viewed as a min–max game between the autoencoder and the discriminator network. The AE consists of an encoder f and a decoder g. Let $x$ be the input feature vector, $z = f(x)$ be the latent variable, $\tilde{x} = g(z)$ be the output of the AE, and $d_z$ be the output of the discriminator network (i.e., the probability that $z$ is sampled from the prior distribution).

- **Reconstruction Loss.** Given the training set $D = \{x_i \mid i = 1, 2, \cdots, M\}$ containing M samples, we first considered the distance between the input data x and its reconstruction $\tilde{x}$. The reconstruction error of each sample is minimized as follows:

$$L_{re}(\omega_f, \omega_g) = \|x_i - g(f(x_i))\|^2, \tag{4}$$

where $\omega_f$ are the encoder weights; $\omega_g$ are the decoder weights; and the $\ell_2$-norm is used to measure the reconstruction errors.

- **Incorporation Loss.** According to the definition of binary cross-entropy loss, we adjusted the weights of the discriminator by minimizing the probability that the latent variable comes from the prior distribution.

$$L(\omega_d) = -\frac{1}{M}\sum_{i=1}^{M}[y_i log\hat{y}_i + (1-y_i)log(1-\hat{y}_i)], \tag{5}$$

where $w_d$ are the discriminator weights; $y_i$ represents the true probability; and $\hat{y}_i$ represents the estimated probability. By setting $\hat{y}_i = 1, i = 1,2,\cdots,M$ and $\hat{y}_i = d_z(g(f(x_i)))$, the expression becomes:

$$L(w_d) = -\frac{1}{M}\sum_{i=1}^{M}log(d_z(g(f(x_i)))), \tag{6}$$

- **Discriminator Loss.** Finally, the discriminator network is trained by minimizing the binary cross-entropy. In more detail, in each iteration, the discriminator is trained on two mini-batches: the first, $x_i$, is sampled from the prior distribution and the second, $z = f(x)$, is the latent variable. The expression of the loss is thus:

$$L(w_g, w_d) = -\frac{1}{2M}\sum_{i=1}^{M}[logd_z(N(0,1)) + log(1-d_z(f(x)))], \tag{7}$$

Note that each small batch can have a number of discriminator and AE iterations greater than 1. The aim of this strategy is to avoid overtraining one of the two networks. In order to stabilize the training process of the adversarial network, a batch normalization algorithm was applied on each layer of the AE and discriminator. The training algorithm for AE is shown in Algorithm 3.

---

**Algorithm 3:** 1DAAE training algorithm.

---

INPUT: $X_{train} = \{x_1, \cdots, x_M\}$ training set divided in M minibatches.

      $K_a$: number of autoencoder training iterations per minibatch

      $K_d$: number of discriminator training iterations per minibatch

      N: number of epochs

OUTPUT: encoder $f$, decoder $g$, discriminator $d$

  $\omega_f, \omega_g, \omega_d \leftarrow$ Initialize parameters for $f$, $g$, $d$

**for** epoch = 1 to N **do**

    **for** i = 1 to M **do**

        $x_i \leftarrow$ i-th minibatch

        **for** k = 1 to $K_a$ **do**

            Update the autoencoder by minimizing the reconstruction error:
$$L_{re} = \|x_i - g(f(x_i))\|^2$$

        **end for**

        Update the autoencoder by minimizing the expression:
$$L(w_d) = -\frac{1}{M}\sum_{i=1}^{M}log(d_z(g(f(x_i))))$$

        **for** k = 1 to $K_d$ **do**

            Update the discriminator by minimizing the binary cross-entropy:
$$L(w_g, w_d) = -\frac{1}{2M}\sum_{i=1}^{M}[logd_z(N(0,1)) + log(1-d_z(g(x)))]$$

        **end for**

    **end for**

**end for**

---

*2.4. DAAE Based on Fault Detection Algorithm*

2.4.1. Two Anomaly Scores Based on the Reconstruction Error and Latent Variables Distribution for 1DAAE

Similar to AEs, the 1DAAE model is trained to learn the description of the normal data. Then, the anomaly scores of each test sample are obtained from the trained 1DAAE model. Samples with high anomaly scores are detected as fault samples.

Due to the unique advantage of the 1DAAE model, we could construct two anomaly scores to enhance the detection performance. The 1DAAE model is composed of encoder $f$ and decoder $g$, which could be used to formulate the anomaly scores to detect fault samples.

For a trained 1DAAE model, the generated samples $\tilde{x} = g(z)$ were similar to the normal samples in the training dataset for any latent variable $z = f(x)$ in the prior latent space (i.e., sample $x$ can be reconstructed perfectly by 1DAAE). However, when $x$ is a fault sample, $\tilde{x} = g(z)$ will have a large reconstruction error with $x$ for any $z$ in the prior latent space.

Therefore, we found two anomaly scores to detect the fault samples cooperatively. One was based on the reconstruction error, and the other was based on the latent variable distribution. For a sample x, the reconstruction error is formulated as in [17]:

$$f_r = \|x - g(f(x))\| \tag{8}$$

where we call the anomaly score an R-score.

The trained discriminator can also formulate an anomaly score. Theoretically, when the 1DAAE reaches the global optimum, the discriminator cannot distinguish between the latent variable subspace and prior distribution in the training dataset. In practice, the discriminator can hardly reach the global optimum. The discriminator is trained with both the latent variable and prior distribution, so it can learn how to distinguish between them. The discriminator based on anomaly score is formulated as

$$f_d = -d_z(f(x)) \tag{9}$$

where $d_z(f(x))$ represents the output of the discriminator for latent variable $z = f(x)$, and the minus is used to make the fault samples have higher anomaly scores than the normal ones. The anomaly score in Equation (9) is called the D-score.

2.4.2. Algorithm of 1DAAE-Based Fault Detection

Based on the above 1DAAE model and two anomaly scores, a novel algorithm of fault detection was proposed. The procedure of the novel fault detection method contains two stages, as shown in Figure 3. In the training stage, the 1DAAE-based fault detection algorithm trains a 1DAAE model on the training dataset $X_{train}$ first by following Subsections 3.1 and 3.2. When the training was finished, parameters in the encoder $f$, decoder $g$, and the discriminator $d_z$ formulated fault scores $f_{dr}$ and $f_d$, respectively, following Equations (8) and (9). Anomaly scores of the training samples were computed and thresholds determined with a certain confidence level. In this paper, the threshold of R-score $T_r$ was determined by 95% of the training samples having scores lower than the threshold as in [17]:

$$T_r = 95 \text{ quantile of } \{ f_r(x) | x \in X\_train \} \tag{10}$$

The decision function on the test dataset is defined as $h_r(x'|X_{train}) = sgn(f_r(x') - T_r)$, where $x' \in X_{realtime}$ is the real-time sample. When the R-score of a sample is higher than the threshold $T_r$, the sample is judged as a fault sample; otherwise, it is considered as a normal sample.

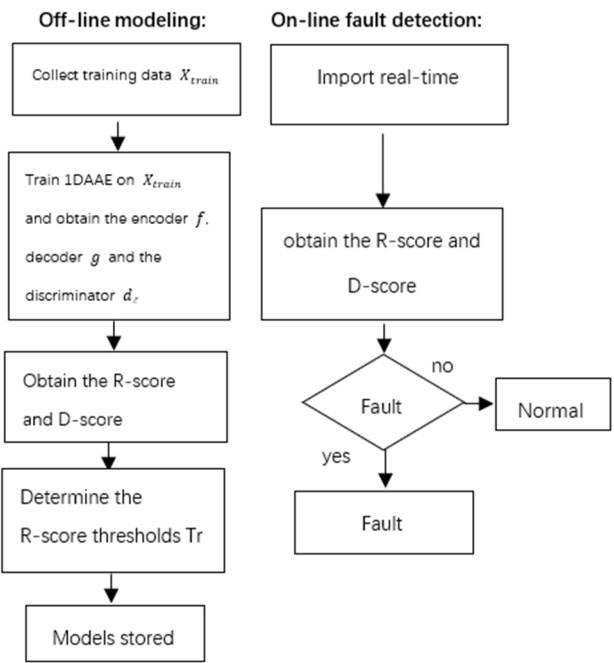

**Figure 3.** Flowchart of the 1DAAE-based fault detection algorithm.

### 3. Results

*3.1. Datasets*

The Tennessee Eastman process (TEP), developed by Down and Vogel [28], is a benchmark simulation problem in chemical engineering. In this experiment, the 1DAAE-based fault detection method was applied to the TEP. The plant shown in Figure 4 consists of five major unit operations: a reactor, a product condenser, a vapor-liquid separator, a recycle compressor, and a product stripper. The process has 12 manipulated variables, 22 continuous process measurements, and 19 compositions sampled less frequently, as shown in Tables 4 and 5. In this study, a total of 33 variables were used for the process monitoring. All composition measurements are included. There were 21 kinds of programmed known faults, as summarized in Table 6. A sampling interval of 3 min was used to collect the simulated data for the training and testing sets. The TEP data can be downloaded from the website address [28]. A total of 960 samples were collected for each fault mode, in which we first let the system run in normal operation and from the 161st sample point, we introduced fault samples. In testing, we first used the normal samples to establish the statistical model including the AE, AAE, 1DAE, GAN, and 1DAAE model, and then we detected 21 kinds of fault data at the same time.

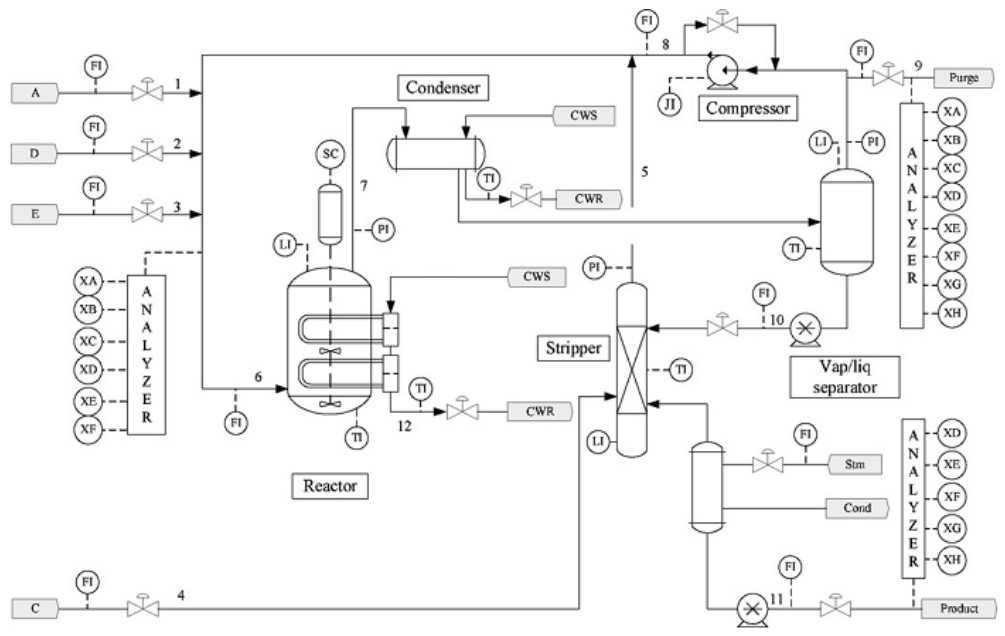

**Figure 4.** Flow sheet of the Tennessee Eastman process.

**Table 4.** Manipulated variables in the Tennessee Eastman process.

| Variable No. | Manipulated Variables |
| --- | --- |
| 1 | D Feed flow valve |
| 2 | E Feed flow valve |
| 3 | A Feed flow valve |
| 4 | A + C Feed flow valve |
| 5 | Recycle valve |
| 6 | Purge valve |
| 7 | Separator valve |
| 8 | Stripper valve |
| 9 | Steam valve |
| 10 | Reactor coolant flow |
| 11 | Condenser coolant flow |
| 12 | Agitator speed |

**Table 5.** Measurement variables in the Tennessee Eastman process.

| Variable No. | Manipulated Variables |
| --- | --- |
| 1 | D Feed rate |
| 2 | E Feed rate |
| 3 | A Feed rate |
| 4 | A + C Feed rate |
| 5 | Recycle flow rate |
| 6 | Reactor feed rate |
| 7 | Reactor pressure |
| 8 | Reactor level |
| 9 | Reactor level |
| 10 | Purge rate |
| 11 | Separator temperature |
| 12 | Agitator speed |
| 13 | Separator pressure |
| 14 | Separator underflow |

| | |
|---|---|
| 15 | Stripper level |
| 16 | Stripper pressure |
| 17 | Stripper underflow |
| 18 | Stripper temperature |
| 19 | Stem flow rate |
| 20 | Compressor work |
| 21 | Reactor coolant temperature |
| 22 | Condenser coolant temperature |
| 23 | Feed % A |
| 24 | Feed % B |
| 25 | Feed % C |
| 26 | Feed % D |
| 27 | Feed % E |
| 28 | Feed % F |
| 29 | Purge % A |
| 30 | Purge % B |
| 31 | Purge % C |
| 32 | Purge % D |
| 33 | Purge % E |
| 34 | Purge % F |
| 35 | Purge % G |
| 36 | Purge % H |
| 37 | Product % D |
| 38 | Product % E |
| 39 | Product % F |
| 40 | Product % G |
| 41 | Product % H |

**Table 6.** Process faults for the Tennessee Eastman process.

| No. | Description | Type |
|---|---|---|
| 1 | A/C feed ratio, B composition constant (stream 4) | Step |
| 2 | B composition, A/C ratio constant (stream 4) | Step |
| 3 | D feed temperature (stream 2) | Step |
| 4 | Reactor cooling water inlet temperature | Step |
| 5 | Condenser cooling water inlet temperature | Step |
| 6 | A feed loss (stream 1) | Step |
| 7 | C header pressure loss—reduced availability (stream 4) | Step |
| 8 | A, B, C feed composition (stream 4) | Random variation |
| 9 | D feed temperature (stream 2) | Random variation |
| 10 | C feed temperature (stream 4) | Random variation |
| 11 | Reactor cooling water inlet temperature | Random variation |
| 12 | Condenser cooling water inlet temperature | Random variation |
| 13 | Reaction kinetics | Slow drift |
| 14 | Reactor cooling water valve | Sticking |
| 15 | Condenser cooling water valve | Sticking |
| 16–20 | Unknown | - |
| 21 | The valve for stream 4 was fixed at the steady-state position | Constant position |

### 3.2. Experimental Results

In this section, we validated the proposed 1DAAE for fault detection. Our experiments were conducted on 21 fault datasets and compared with three fault detection methods (i.e., AE, AAE, 1DAE, and GAN). Our approach was implemented in PyTorch (1.2.0 with python 3.7) by optimizing the networks using the SGD optimizer with an initial learning rate lr = 0.001, weight decay 0.0005, and momentum 0.9. We trained our model with a batch size of 100 on the normal dataset. Additionally, all our experiments were executed on a PC with an Intel(R) Core (TM) i7–7700 3/4 GHz processor, 8GB RAM.

To show the effectiveness of our method, some parameter settings were designed. The TEP dataset was taken as the example for the parameter analysis. We studied three aspects that influenced the performance of 1DAAE in the anomaly detection task (i.e., the number of epochs, latent vector size, and anomaly score).

The main drawback of the 1DAAE is the instability during training. The balance between the AE and the discriminator is important. The training process of 1DAAE is an EM (expectation-maximum) process, that is, first we train the AE and then train the discriminator, and then reciprocate. The influence of the number of epochs of 1DAAE is to balance the AE and discriminator, so the loss trend of the AE and discriminator is stable, and the 1DAAE training process is stable.

The number of epochs in 1DAAE. We present the trend of the loss of AE with respect to (w.r.t.) varying values of the number of epochs. The number of epochs is from 1 to 500, and the trend of the loss of AE is shown in Figure 5, where the blue poly-line represents the trend of the loss of AE and the yellow straight line represents the ordinate of 0.03. The loss of AE is calculated according to Equation (3), which represents the error of the auto-encoder, and we want it to be as small as possible. It can be seen in Figure 5 that when the number of epochs exceeded 300, the loss of AE approached 0.03.

We also present the trend of the loss of discriminator w.r.t. varying values of the number of epochs. The number of epochs goes from 1 to 500, the loss of discriminator is shown in Figure 6, where the blue poly-line represents the trend of the loss of discriminator and the yellow straight line represents the ordinate of 0.5. The loss of discriminator is calculated according to Equation (4), which represents how well the latent variable z fits with the prior distribution. If the latent variable z fits with the prior distribution well, the discriminator has no ability to tell whether the sample is sampled from the prior distribution or from the latent variables. Therefore, we want the loss of the discriminator to be equal to 0.5 approximately, which is the line shown in Figure 6. Considering Figures 5 and 6, we chose the number of epochs as 400, and at this time, the loss of AE is 0.027 and the loss of the discriminator is 0.498.

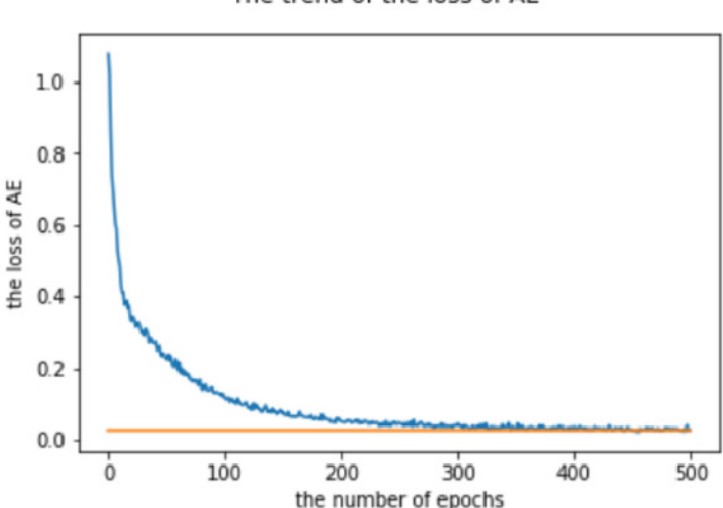

The trend of the loss of AE

**Figure 5.** The trend of the loss of AE w.r.t. varying values of the number of epochs.

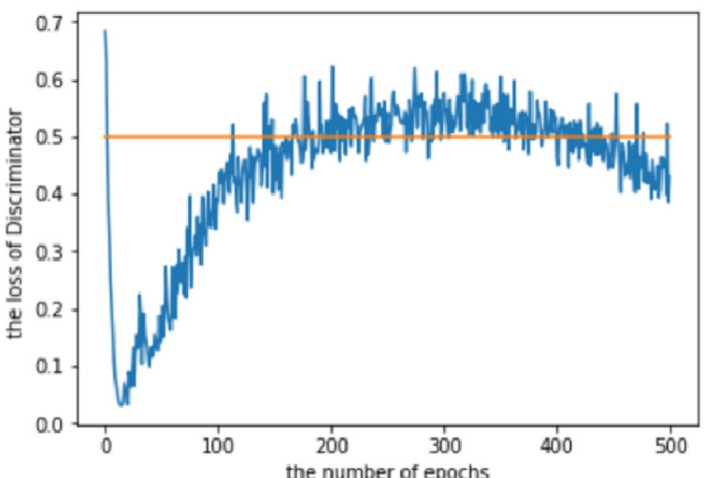

**Figure 6.** The trend of the loss of discriminator w.r.t. varying values of the number of epochs.

Latent vector size in 1DAAE. We present the trend of the mean accuracy w.r.t. varying values of latent vector size, which is in a range of {5, 10, 15, 30, 40, 50}. The mean fault detection results are shown in Figure 7, where the lower figure of the blue bar represents the late vector size, and the upper figure represents the mean fault detection result obtained by using the lower late vector size. We can see that the results showed an approximate normal distribution and the best mean fault detection result was 0.86 when we chose 15 as the latent vector size.

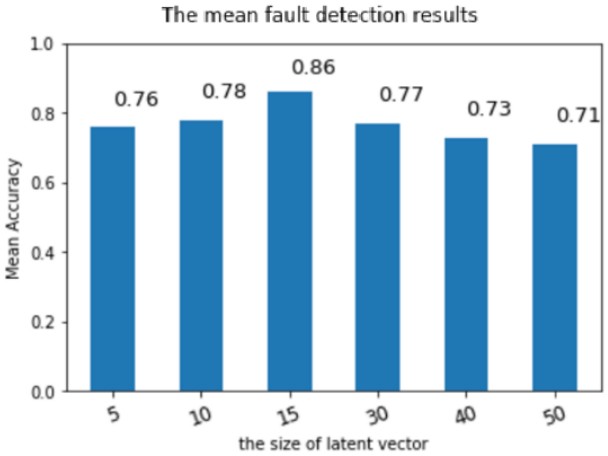

**Figure 7.** The mean fault detection results w.r.t. varying values of the latent vector size.

Two anomaly scores in 1DAAE. Anomaly scores are the crucial component that directly influences the performance of fault detection methods. Two types of anomaly scores are proposed in Section 2: the R-score and D-score. The R-score is calculated according to Equation (8), which is based on the reconstruction error. We hope that the R-score of the normal sample is lower than the threshold Tr, while the R-score of the fault sample is higher than the threshold Tr. The Tr is calculated according to Equation (10) in Section 2. The D-score is calculated according to Equation (9), which is based on latent vector distribution. We hope that the latent vector of the normal sample will fit with the prior distri-

bution, while the latent vector of the fault sample should stay away from the prior distribution. To show the results of the R-score and D-score, we chose 2 as the latent vector size. The R-score of the 1DAAE for fault 1 is shown in Figure 8, where the blue poly-line represents the R-scores of samples and the yellow straight line represents the Tr. We can see that samples from 1 to 160 are normal samples whose R-scores are lower than Tr and the samples from 161 to 960 are fault samples whose R-scores are higher than Tr. It is a good result for the R-score of the 1DAAE. The R-scores of the AE, AAE, and 1DAE for fault 1 are shown in Figures 9–11, where the blue poly-lines represent the R-scores of the samples and the yellow straight lines represent the Tr. We can see that their R-score trends were almost similar. This may be because, first, fault 1 is relatively simple to detect and second, these algorithms have similar encoder and decoder network structures to calculate the R-scores. The difference lies in the value of R-scores for these algorithms. Among them, the R-scores of AE and AAE were less than those of the 1DAE and 1DAAE, which may be due to the fact that the 1D convolution layer extracts more distinctive features between the normal and fault samples.

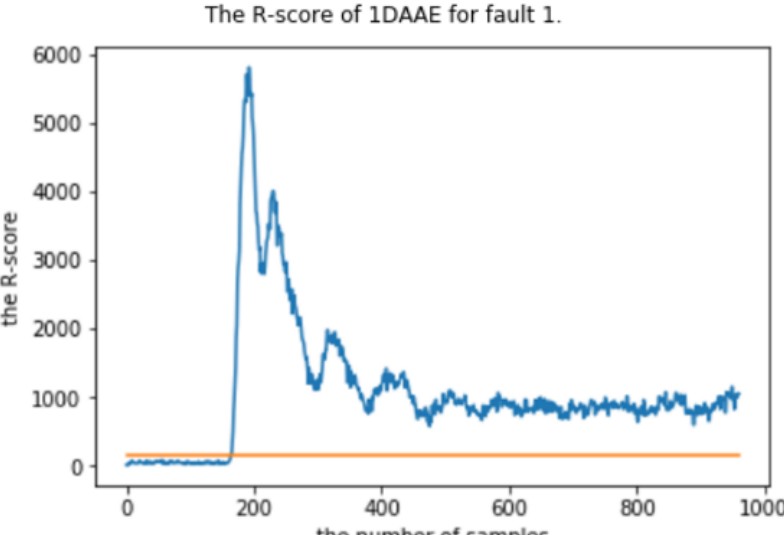

**Figure 8.** The R-score of the 1DAAE for fault 1.

**Figure 9.** The R-score of the AE for fault 1.

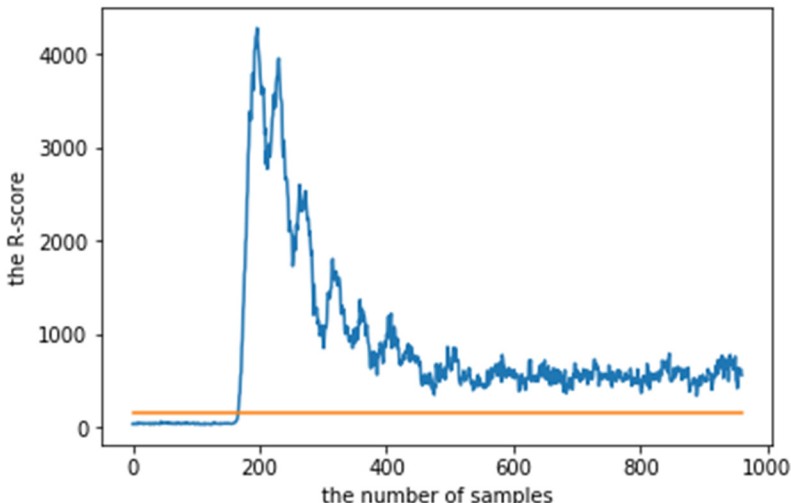

**Figure 10.** The R-score of the AAE for fault 1.

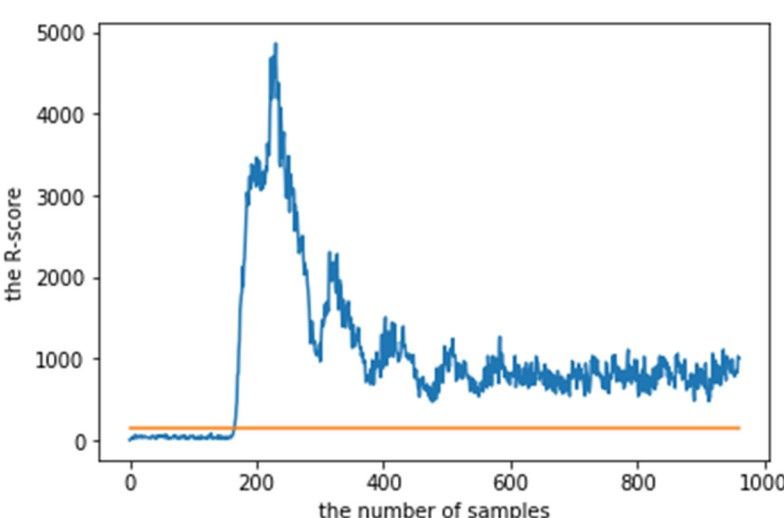

**Figure 11.** The R-score of the 1DAE for fault 1.

The latent vector z of 1DAAE for fault 1 is shown in Figure 12. We can see that the points *
represent normal samples that fit with the prior distribution. The point x represents fault samples
that stay away from the prior distribution. It also showed a good result for the D-score. The latent
vector z of AAE for fault 1 is shown in Figure 13. We can see that the D-score is helpful for distin-
guishing between the normal and fault samples. It should be noted that only AAE and 1DAAE
could calculate D-scores in this paper and the training of GAN was carried out according to [27].

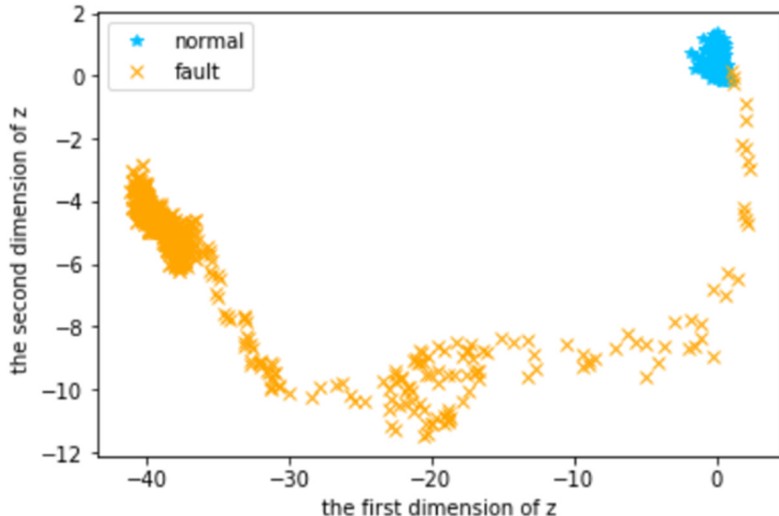

**Figure 12.** The latent vector z of the 1DAAE for fault 1.

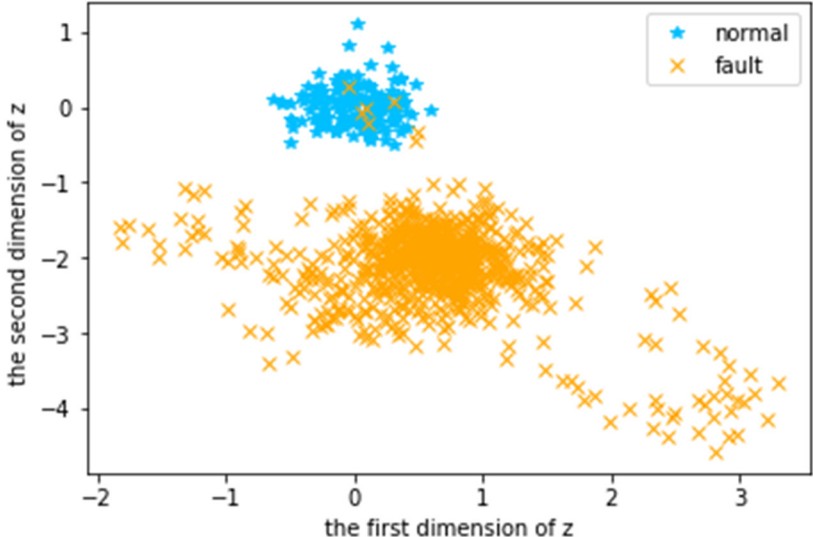

**Figure 13.** The latent vector z of the AAE for fault 1.

After determining the optimal parameters, 1DAAE was modeled according to the network architecture mentioned in Section 2 and was trained according to Algorithm 3, the 1DAAE training algorithm. In comparison algorithms, the network architecture of the AE consisted of an encoder and decoder. The network architecture of the encoder and decoder was similar to 1DAAE, but removed a 1D convolution layer; the training process of the AE was according to Algorithm 1, the autoencoder training algorithm. The network architecture of 1DAE consisted of an encoder and decoder. The network architecture of the encoder and decoder was similar to 1DAAE, and the training process of 1DAE was according to Algorithm 1, the autoencoder training algorithm. The network architecture of the AAE consists of the encoder, decoder, and discriminator. The network architecture of the encoder and decoder was similar to 1DAAE, but removed the 1D convolution layer; the training process of AAE according to Algorithm 3, the 1DAAE training algorithm. Because the models and training process were similar, the parameter settings were almost the same for the AE, 1DAE, AAE, and 1DAAE. The network architecture and training of GAN were carried out according to [27].

In fault detection, we detected 21 known faults in TEP based on the AE, AAE, 1DAE, GAN, and the proposed 1DAAE method. The results are shown in Figure 14. For each fault, five histograms represent the accuracy of the above five methods. The dark golden-rod histograms represent the accuracy of AE; the sea-green histograms represent the accuracy of AAE; the blue histograms represent the accuracy of the 1DAE; the red histograms represent the accuracy of 1DAAE; and the gray histograms represent the accuracy of GAN. We can see that all four methods offered high accuracy for fault numbers (1–2), (4–8), (12–14), and (17–18), where the difference between the highest and lowest accuracy was between 0.03. These faults can be linearly distinguished. We found that the AE achieved the highest accuracy among them, which is possibly because the linear method has the ability to handle them better. The other methods still offered good results, and there was only a small difference between them. For fault numbers (3), (9), (10), (15-16), and (19), 1DAAE provided the best fault detection performance over all the other methods; for fault numbers (11), (20), and (21), 1DAAE provided the second best accuracy. These faults were nonlinear and difficult to distinguish. Especially for fault numbers (3), (9), and (15), we could see that the AE provided low accuracy, and thus could not detect the faults successfully. For fault numbers (11), (20), and (21), GAN provided the best fault detection performance, however, for fault numbers (3), (9), (15), and (19), GAN provided the worst fault detection performance. We can see from Figure 15 that GAN obtained the lowest average accuracy, so the performance of GAN was not stable enough. The performance of AAE was better than AE, possibly because the D-score is advantageous to the improvement in the fault detection performance. We also found that the performance of 1DAAE was better than 1DAE, which confirms our hypothesis. The performances of 1DAAE and 1DAE were better than AE and AAE, which illustrates that the one-dimension convolutional layers used in our methods have the ability to extract more valuable features for fault detection, and the one-dimension convolutional layers were more advantageous for fault detection than the D-score. Finally, the mean accuracy based on the AE, AAE, 1DAE, GAN, and 1DAAE is shown in Figure 11. We found that 1DAAE offered the best performance with the help of the one-dimension convolutional layers and D-score.

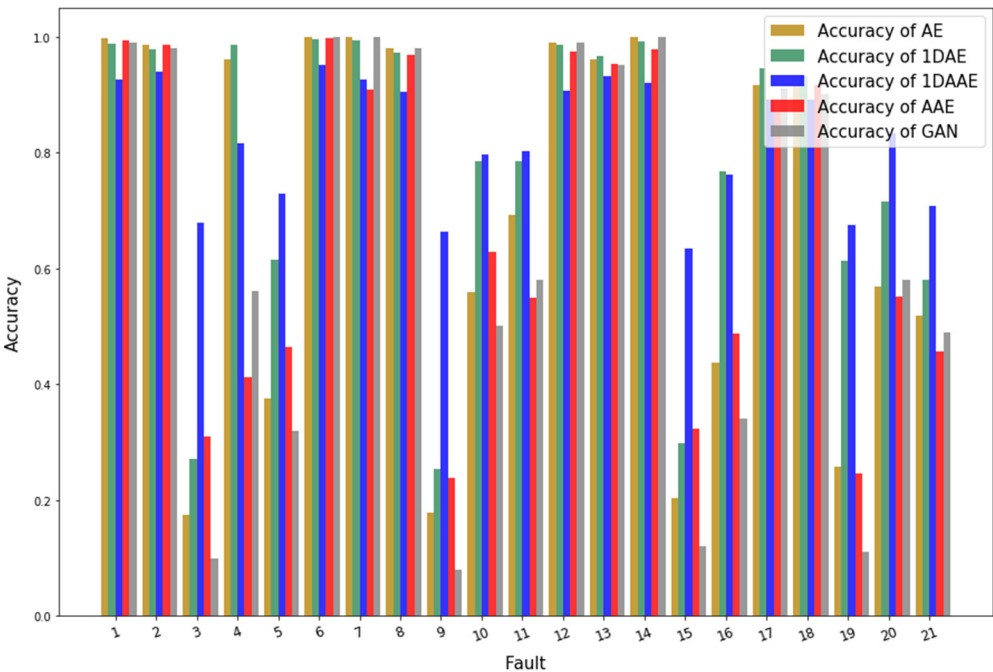

**Figure 14.** The fault detection results based on the AE, AAE, 1DAE, GAN, and 1DAAE.

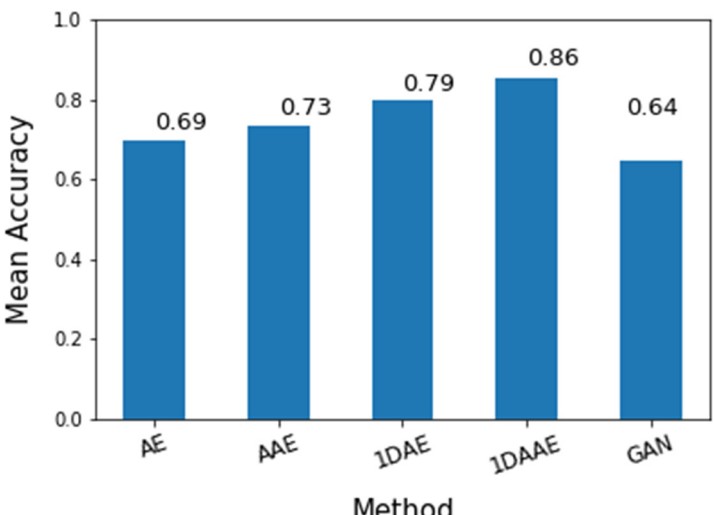

**Figure 15.** The mean accuracy based on the AE, AAE, 1DAE, GAN, and 1DAAE.

**4. Conclusions**

In this paper, we proposed a novel unsupervised fault detection method named 1DAAE, which introduced two new ideas: 1D convolution layers for the encoder to obtain better features and the adversarial thought, which is to impose the latent variable z to cluster into a prior distribution. Then, two anomaly scores, R-score and D-score, for 1DAAE were proposed to detect the fault samples, one based on reconstruction errors, and the other based on the latent variable distribution. Extensive experiments conducted on TEP prove the effectiveness of our methods. Through the experiments, we found that the both the 1D convolution layers and the latent vector distribution are helpful for fault detection, and 1D convolution layers are more helpful. Future work will consider more feature extraction techniques applied to AE.

**Author Contributions:** Conceptualization, J.W. and Y.L.; Methodology, J.W.; Software, J.W. and Y.L.; Validation, J.W. and Y.L.; Writing—review and editing, J.W., Z.H. and Y.L.; Visualization, J.W. All authors have read and agreed to the published version of the manuscript.

**Funding:** This research was funded by the National Nature Science Foundation (61503038, 61403042); the Scientific research project of Education Department of Liaoning Province (LQ2020013, LJKMZ20221484); a grant from the Bohai University Teaching Reform Program (No. YJG20210023); a grant from the Ministry of Education industry-University Cooperative Education Program (202102599009, 202101332004, 202101337001,220504643183656); and the Application Basic Research Plan of Liaoning Province (2022JH2/101300282).

**Data Availability Statement:** Not applicable.

**Conflicts of Interest:** The authors declare no conflicts of interest.

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
