# Peer review of "A Novel Fault Detection Method Based on One-Dimension Convolutional Adversarial Autoencoder (1DAAE)"

_processes, doi:10.3390/pr11020384_

Round 1
Reviewer 1 Report
In this article, “A Novel Fault Detection Method Based on One Dimension Convolutional Adversarial Autoencoder (1DAAE). The topic is very important, but I think that this contribution does not rise to become a full contribution to your journal, so my advice I recommend improving the and give the clear description in terms of training process learning for normal and fault. However, I have some comments and suggestions for the authors. as follows:
1- There are shortcomings in the abstract, so it should be clarified the contribution in this section. so no need to putted these sentences inside the abstract "The network architectures of 1DAAE include three parts: the encoder, the decoder, and the discriminator for latent variables. The encoder tries to learn the latent variables subspace from the original operational data, the decoder tries to reconstruct the operational data from the latent variables subspace."
2- Must write the whole word for the abbreviations such as " T2".
3- The writing of this letter is difficult to follow. For example, the full names are not given for too many abbreviations.
4- The writing in line (46) is not good, so must paraphrase it again.
5- In section material and method very lack for giving a clear expression for "?(?) = ?(?|?)?(?)".
6- The system model lacks a description of the “Autoencoder training algorithm”, “the training process for autoencoder, and the discriminator network" modeling process.
7- How to obtain the data to evaluate the results? The reviewer does not think data from software can work for the actual environment.
8- There is a lack of comparison algorithms in the simulation experiment for the training process for "normal and fault"
9- Must write the subtitle or labels in the section simulation results from (Fig. 5 to Fig. 8)
10- Must improve the analysis of the proposed method, simulations were performed via multiple aspects. It is encouraged to include some more recent works as the baseline.
Reviewer 2 Report
Aiming at the problem that traditional fault detection methods based on automatic encoder (AE) ignore a large number of useful information about the distribution of potential variables, this paper proposes a new unsupervised fault detection method called one-dimensional convolution adductive autoencoder (1DAAE), which introduces two new ideas: one-dimensional convolution layer of encoder to obtain better features, And thought the potential variables ��� clustering into prior distribution. This method not only has a stronger feature representation ability than traditional AE, but also enhances the recognition ability by introducing the prior distribution of latent variables into clustering.
The research method of this article is rational, the writing format is standardized, But the following problems exist.
(1)In the abstract part, the introduction of the purpose of this study is not clear enough, and the description of the experimental results is lacking. This paper focuses on unsupervised learning in fault detection, and suggests that unsupervised learning should be used as a key word.
(2)In the introduction part, this paper introduces some current research status, but lacks a summary. It is suggested to make a certain summary of the mentioned research methods and theories.
(3)In the second part of this paper, we can reduce the introduction of the mature automatic encoder, and increase the introduction of the new method and its advantages.
(4)In the third part of this paper, the influence of the number of epoids on the performance of 1DAAE in the anomaly detection task is studied by the loss trend of AE and discriminator, but the influence of the loss trend of both on the performance of the new method 1DAAE is not explained.
Round 2
Reviewer 1 Report
I think the authors didn't answer some of our comments, so I suggest to revised again:
1- There is a lack of comparison algorithms in the simulation experiment for the training process for "normal and fault"
2- The authors must write the subtitle or labels in the section simulation results from (Fig. 5 to Fig. 8).
3- the authors must support the citation reference for both equations Eq. (8) and Eq.(10).
4- Must delete the self-citations by authors.
